# Surface Evaluation of Orthodontic Brackets Using Texture and Fractal Dimension Analysis

**DOI:** 10.3390/ma15062071

**Published:** 2022-03-11

**Authors:** Michał Sarul, Marcin Mikulewicz, Marcin Kozakiewicz, Kamil Jurczyszyn

**Affiliations:** 1Department of Integrated Dentistry, Wroclaw Medical University, Krakowska 26, 50-425 Wroclaw, Poland; 2Department of Dentofacial Orthopedics and Orthodontics, Division of Facial Abnormalities, Wroclaw Medical University, Krakowska 26, 50-425 Wroclaw, Poland; marcin.mikulewicz@umw.edu.pl; 3Department of the Maxillofacial Surgery, Medical University of Lodz, 113 S. Żeromski Street, 90-549 Lodz, Poland; marcin.kozakiewicz@umed.lodz.pl; 4Department of Dental Surgery, Wroclaw Medical University, Krakowska 26, 50-425 Wroclaw, Poland; kamil.jurczyszyn@umw.edu.pl

**Keywords:** orthodontic wire, orthodontic wire surface, fractal dimension analysis, texture analysis

## Abstract

The surface topography of orthodontic brackets can have a significant impact on both the effectiveness of the therapy and the behavior of these elements in the oral cavity environment. In this situation, striving to obtain the most uniform, smooth surface in a repeatable manner for each manufactured element should be a sine qua non condition for each supplier of orthodontic brackets. Therefore, it is necessary to analyze the surfaces of orthodontic brackets using different methods. One of them—that is relatively simple and repeatable—is the analysis of the fractal dimension and the analysis of the textures of the optical images on the surface. In the presented study, fractal dimension analysis and texture analysis were performed by selecting four brackets from three different manufacturers (Mini Sprint, Sprint, Nu-Edge, Orthos SS). The area of each bracket slot was analyzed at six predefined points. The smoothest and most uniform and reproducible surface structure was shown by the Mini Sprint bracket. On the other hand, Sprint brackets showed the least homogeneous and least repeatable surface structure.

## 1. Introduction

Fixed appliances have been one of the main tools used in orthodontic therapy for over a century. The effectiveness of such a therapy largely depends on the mechanical properties of the components of an appliance. Orthodontic brackets are the elements permanently attached to the teeth, meaning that they are involved throughout the whole treatment period. Their mechanical features affect interaction with an orthodontic wire, thus determining the performance of the entire appliance and ultimately the effectiveness of the treatment. That is why testing and improving the properties of orthodontic brackets has been the subject of research for many years [1].

Orthodontic brackets are manufactured from different materials: stainless steel, titanium, monocrystalline ceramics, polycrystalline ceramics, polymers [2].

Brackets made of monocrystalline ceramics offer the greatest smoothness of the surface and very high hardness. Hardness can be a big advantage, however it requires careful workmanship and planning of the rounded shape of the edges of the bracket so as not to cause a notching phenomenon on the wire surface. Polycrystalline brackets show very high roughness. Both types of ceramic brackets are easily cracked and cause high resistance to motion when moving the bracket along the wire. Polymer brackets have a much lower hardness than ceramic ones, so they can be gradually rubbed off during treatment. In addition, usually they create a lot of friction in the orthodontic archwire-bracket space. Metal brackets made of titanium or stainless steel are the most widely used. Titanium brackets are gaining popularity due to their high biocompatibility. Their disadvantage is the possibility of cold welding with beta-titanium wires and usually have slightly higher roughness compared to stainless steel brackets. Steel brackets are made of Cr-Ni 18-8 stainless steel. They are the most popular type and offer a high smoothness with relatively low resistance to motion, as well as hardness close to the parameters of the orthodontic wires used [1,2].

The most common manufacturing methods of metal brackets are casting or molding. It is less common to manufacture orthodontic brackets by laser welding of the base and wings of the bracket, each of which is made using a separate method.

The surface properties of the orthodontic brackets’ slot may affect the treatment process, especially in terms of the friction generated in the archwire-slot space, bacteria adhesion and ion release [3,4]. Unfortunately, aspects of mechanical properties of the orthodontic bracket slot are neglected in many original studies [3]. The surface structure depends, to the greatest extent, on the accuracy of the vendor’s manufacturing of the bracket. An extremely important aspect of the production process is not only the need to obtain the smoothest possible slot surface but also to maintain the repeatability of the process. The results of all material tests allow for predicting the course of mechanical processes between the bracket and the wire, but only when we assume that all brackets (and wires) of a given manufacturer obtain repeatable mechanical properties in the production process, including the surface structure.

As a relatively simple technique that provides information about the complexity of the geometric structure of a surface, computer image analysis and the quantity, such as fractal dimension, can be used. Fractal and multifractal surface properties have so far been determined for many materials, including metallic materials and their alloys, ceramic, polymeric and amorphous materials [5,6,7,8,9,10,11,12,13]. With this in mind, it can be assumed that the fractal analysis method can be a reliable and relatively simple method that may be used to compare the complexity and repeatability of slot surfaces of different orthodontic brackets.

The aim of this study was to assess the homogeneity of the surface of the slots of unused brackets from different manufacturers by analyzing textures and fractal dimensions.

## 2. Materials and Methods

The null hypothesis was that all brackets from one manufacturer exhibit a homogeneous surface structure, as well as that brackets from different manufacturers show no differences in surface structure.

Brackets from the following manufacturers were selected for the study:

Forestadent—Sprint II and Mini Sprint II;

TP Orthodontics—Nu-Edge;

Ormco—Orthos SS.

The study group includes central incisor brackets, 2 right and 2 left, with a slot dimension of 0.022 × 0.025 inches (width × depth).

Each bracket was marked as follows:

Mini Sprint II—MS;

Sprint II—S;

Nu-Edge—N;

Orthos SS—O.

On each of the brackets, the surface of the bottom of the slot was examined at 6 points (Figure 1), meaning that 24 measurement points were obtained for brackets of a given manufacturer:

### 2.1. Taking Images

All images were taken using the scanning electron microscope (SEM)—VEGA3 (Tescan, Brno—Kohoutovice, Czech Republic). The resolution of images was 1280 × 1430, with a magnification of 276×, a voltage of 30 kV, and a backscattered electron (BSE) detector. On all the slots’ surface images, six regions of interest (ROIs) for fractal dimension and texture analyses were set. All ROIs were 200 mm × 200 mm in size. All images were saved as 8-bit grayscale bitmaps. All graphic operations were performed using GIMP version 2.10.30 (GNU Image Manipulation program: www.gimp.org, free and open source license, accessed on 21 January 2022).

### 2.2. Fractal Dimension Analysis

All fractal analyses were performed in ImageJ, version 1.53e (Image Processing and Analysis in Java—Wayne Rasband and contributors, National Institutes of Health, USA, public domain license, https://imagej.nih.gov/ij/ accessed on 1 January 2022), and the FracLac plugin, version 2.5 (Charles Sturt University, Australia, public domain license).

In our study, we decided to use a modified algorithm of the box-counting method, which makes it possible to analyze monochromatic images, such as 8- or 16-bit images. In the case of grayscale images, we applied the intensity difference algorithm to calculate the fractal dimension. This algorithm is fully described in our previous study [14,15]. The analyzed image is divided into boxes, as in the box-counting method. The image size selected for analysis was 200 × 200 mm. FDA consists of some repeatable steps, for example: in the first step, gird size equals 200 mm (dimension of analyzed image, ε = 1), in the next steps ε is divided by 2 (ε value for following steps: ε = 0.5, ε = 0.25). In each step, the difference of pixels’ bright intensity is calculated in every gird on scale ε. In the FracLac plugin, the algorithm of the ε calculation is called block series. This option scans a square block within an image using a series of grids calculated from the block size. This specific way is most usable for the analysis of a pattern which fills the whole area of the image.

The difference between the maximum pixel intensity and the minimum pixel intensity is calculated in each box (δI_i,j,ε_, where i, j—the location of the analyzed box in the ε scale):δI_i,j,ε_ = maximum pixel intensity_i,j,ε_ − minimum pixel intensity_i,j,ε_
(1)

In the next step, 1 is added to the intensity difference to prevent its value from becoming a 0:I_i,j,ε_= δI_i,j,ε_ + 1(2)

Finally, the fractal dimension of the intensity difference is described using the following formula:(3)$$\mathrm{FD}=\underset{\varepsilon \to 0}{\lim}\frac{\ln\left(I\varepsilon \right)}{\ln\left(\frac{1}{\varepsilon }\right)}$$
where FD is the final fractal dimension of intensity, Iε = Σ [1δI_i,j,ε_ + 1], ε is box scale.

### 2.3. Texture Analysis

The surface texture of orthodontic brackets was evaluated using features derived from two groups (run-length matrix and co-occurrence matrix) and the previously described Texture Index (TI) [12,16]. The regions of interest (ROIs) were normalized (μ ± 3σ) to share the same average (μ) and standard deviation (σ) of optical density within the ROIs. The selected image texture features (entropy and difference entropy from the co-occurrence matrix and long-run emphasis moment from the run-length matrix) in ROIs were calculated:(4)$$\mathrm{Entropy}=-\sum_{i=1}^{\mathrm{Ng}}\sum_{j=1}^{\mathrm{Ng}}p\left(i,j\right)\log$$
where Σ is the sum, Ng is the number of optical density levels in the radiograph, i and j are the optical density of pixels that are 5 pixels away from one another, p is probability, and log is the common logarithm [13],
(5)$$\mathrm{LngREmph}=\frac{\sum_{i=1}^{\mathrm{Ng}}\sum_{k=1}^{\mathrm{Nr}}k^{2}p\left(i,k\right)}{\sum_{i=1}^{\mathrm{Ng}}\sum_{k=1}^{\mathrm{Nr}}p\left(i,k\right)}$$
where Σ is the sum, Nr is the number of series of pixels with density level i and length k, Ng is the number of levels for image optical density, Nr is the number of pixels in the series, and p is probability [17,18,19,20,21]. Long-run-length emphasis moment (LngREmph) was computed from data taken from the bracket surface visualized in SEM, and measures of disarrangement (Entropy) were computed as non-directional measures. The two equations given above were subsequently used for the Texture Index construction [11]. Finally, the Texture Index (TI), which represents the ratio of the measure of the diversity of the structure observed in the images to the measure of the presence of uniform longitudinal structures, was calculated:(6)Texture Index=EntropyLngREmph=(−∑i=1Ng∑j=1Ngp(i,j)log(p(i,j)))∑i=1Ng∑k=1Nrp(i,k)∑i=1Ng∑k=1Nrk2p(i,k)

### 2.4. Statistical Analysis

Statistica, version 13.3 (StatSoft, Cracow, Poland) was used to perform statistical tests in the aspect of fractal dimension analysis. A value of 0.05 was deemed to be statistically significant. The Shapiro–Wilk test was applied to confirm the normality of distribution. Due to the normal distribution, parametric tests were performed. The analysis of variance (ANOVA) and the least significant difference post hoc were applied to reveal the fractal dimension differences.

Texture comparisons between wire sides and material were performed with the one-way ANOVA or the Kruskal–Wallis test, depending on the presence of normal distribution. Simple regression analysis was also done to investigate relationships between general mineral condition parameters and radiological texture features. When *p* < 0.05, the difference was considered statistically significant. Statgraphics Centurion 18, version 18.1.12 (StarPoint Technologies, Inc., Falls Church, VA, USA), was used for statistical analyses.

## 3. Results

The average values of the fractal dimension of slot surfaces for each series are shown in Table 1. Analysis of variance showed that the fractal dimension values of slot surfaces of different series differed only in the S brackets (*p* = 0.00015). For the other brackets, there were no statistical differences between the FD for the slot surface of each series (*p* < 0.05). Fully detailed results containing the p value of ANOVA post hoc test are shown in the Appendix A.

The results of the analysis of variance with a post hoc least significant difference test for the values of the fractal dimension of the surface of the slots from each series are presented in Table 2. There was a statistical difference in the mean value of the fractal dimension of the surface of the S bracket slots compared to the other types of tested brackets. There were no statistical differences in the mean FD value between MS, O and N bracket slots. The lowest value of the fractal dimension of the slot surface was recorded for MS brackets and was 1.6431. The highest FD value was observed in S bracket slots. It is worth noting that the lowest SD value was recorded for O brackets (SD = 0.0218), while the highest was for the S and N brackets (SD = 0.0389). Fully detailed results containing the p value of ANOVA post hoc test are shown in the Appendix A.

The Texture Index is lowest in MS brackets (*p* < 0.0001). The other groups presented similar TI values, i.e., there were no statistically significant differences (Figure 2 and Figure 3).

According to the explanation of the TI meaning, which was presented by Sarul et al. [15], higher values of TI describe smoother surfaces, i.e., the friction is lower in contact with that surface.

## 4. Discussion

In classical Euclidian geometry, we used to know that dimension is an integer value. This value indicates how many variables we need to describe a dimension of examined object. For example, 0 is the dimension of point; to describe a segment of line, we need 1 variable-length, flat figures have 2 dimensions (length, width) and solids have 3 dimensions (length, width, height). In fractal geometry, the number of dimensions is the rational number in the range between 0 and 3. For example, FD of square is 2, Sierpinski’s carpet equals approximately 1.8928. It means that in the infinity scale, Sierpinski’s carpet is something between the 1 dimension and 2 dimensions’ shape with a tendency to 2 dimensions (1.8928 is closer to 2 than 1). The fractal dimension value of images in finite scale (for example microscopic photography) becomes a value between 1 and 2. The lower the value of FD, the more complex is the analyzed shape.

Among all samples, all brackets showed a fractal dimension closest to 1.7, and thus the highest proportion of the heterogeneity of the surface-type structure. O brackets showed the smallest standard deviation (SD = 0.0218). The other brackets showed a very similar and also small standard deviation (SD = 0.0389–0.0335). The results obtained reveal that, in general, the tested brackets present a uniform surface structure along the entire bracket slot. Furthermore, brackets from one manufacturer do not differ significantly in this respect; this is significant because the fractal dimension makes it possible to express the nature of the surface disturbances that occur numerically. A significant difference in the fractal dimension value would indicate that there is either a linear or surface/two-dimensional roughness on the surface of the brackets. The presented study showed that most of the brackets are made with a uniform surface structure with irregularities of the surface/two-dimensional type. Only S brackets did not show a surface with uniform features when comparing different measurement points along one bracket slot as well as between brackets. The results obtained clearly demonstrate that the manufacturing process of metal orthodontic brackets does not always allow for a relatively good unification to be achieved in terms of surface characteristics. Fractal analysis has shown the S brackets as the most heterogeneous in this respect. On the other hand, the significant variation in the texture of the slot surface in the N- and O-series brackets may also be due to insufficient quality control during the manufacturing process. Here, the manufacturing process made it impossible to obtain a product adequately unified in terms of the structure of the slot surface, i.e., the part of it which was most responsible for the interaction with the surface of the orthodontic wire.

Orthodontic treatment that uses sliding mechanics involves strong mechanical interaction between two surfaces—the surface of the orthodontic wire and the surface of the bracket slot. The properties of these surfaces can significantly affect the course of this interaction.

It is important to consider whether the observed differences may have clinical relevance in orthodontic therapy with fixed braces. The topography of the bracket slot surface may influence the following factors: friction generated in the orthodontic arch/bracket system, bacterial adhesion and ion release [22,23,24,25,26]. D’Antò et al. postulated that the roughness of orthodontic brackets could affect friction in the arch-bracket system. These assumptions have been confirmed by many researchers, including Doshi et al. The latter confirmed that friction increases with an increasing roughness of both orthodontic wires and brackets [25,27]. Raji et al. and Oliveira et al. drew attention to the problem of the influence of the surface structure of the elements of fixed appliances on the degree of bacterial biofilm formation and its influence on the possible complications during orthodontic treatment [28,29]. Eliades first raised the issue of the influence of the surface structure on the degree of metal ion release and the destructive effect of corrosion on the elements of fixed appliances in the oral cavity. The high degree of correlation between the roughness and the severity of the corrosive phenomena was then investigated and confirmed by Nalbantgil et al. and Shin et al. [30,31,32].

The friction value is one of the most important factors affecting the biomechanics of tooth sliding in sliding mechanics. Much of the consideration of both tooth alignment and the distalisation phases concerns the effect of the friction coefficient on the efficiency of this movement [22,23,24,25]. The topography of the bracket surface is one of the factors that can greatly affect the value of the friction coefficient. Nevertheless, this aspect is often overlooked in many publications [3]. Studies focusing on evaluating the surface of the bracket slot usually apply a relatively complex profilometry test or AFM (atomic force microcopy) analysis for this purpose [25,26,27]. Fractal dimension analysis is a relatively simple method that allows the surface variation of brackets to be assessed by mathematically analyzing a digital image obtained from an optical microscope or SEM. The results obtained by the authors presented a relatively large variation in the surface of metal brackets—between samples from the same manufacturer as well as between brackets from different manufacturers. This is consistent with the results obtained by Agarwal et al., Park et al., and Lee et al., who also showed a large variation in the surface roughness of brackets from different manufacturers [25,26,27]. Furthermore, these authors have also indicated that a large standard deviation in the degree of surface roughness can apply to the slot of a single bracket. The occurrence of a large standard deviation, in terms of the variation in surface topography between the brackets from the same manufacturer or even within the surface of a single slot of each bracket, does not really allow the results obtained in profilometry tests to be extrapolated to a clinical reality. This study proves that the lack of standardization of manufacturing methods, in this aspect, means that the friction coefficient of some brackets cannot be considered predictable. In this respect, in fractal analysis, all brackets except S displayed a high degree of uniformity. In contrast, texture analysis showed a high degree of surface inhomogeneity for O and N brackets. Of the orthodontic bracket surfaces tested here, MS brackets have the highest TI value, so it is to be expected that the friction in these brackets will also be the lowest among all elements compared. Furthermore, it should be stated that the bracket surfaces tested here generally have a higher TI than those found on orthodontic arches [15], meaning the slot surfaces of the tested brackets are generally smoother than those of orthodontic arches.

Studies conducted so far have proven that the adhesion of pathogenic microorganisms depends, among other things, on the degree of surface roughness of the components used in the oral cavity [28,29,30]. In relation to orthodontic brackets, studies have been conducted which have shown various bacterial adhesion depending on the type of bracket (conventional/self-ligaturing) and the material from which it is made [11,30,31,32]. Furthermore, there have been studies on orthodontic wires that have proven a link between biofilm formation and the degree of roughness [27,33,34,35]. Tawfik et al. proved that there is a strong, linear correlation between the roughness of orthodontic archwires and the formation of bacterial biofilm [35]. There are no such studies on brackets, however, one should expect a similar correlation for metal brackets. The presented experiment proved that, when examined by fractal analysis, brackets that are made of the same material can differ significantly in terms of surface topography. More importantly, however, they can also differ in this respect in terms of brackets from the same manufacturer. Considering fractal analysis, this was the case for S brackets, while texture analysis showed significant heterogeneity of O and N brackets. Texture analysis showed that MS brackets presented the smoothest surface, thus, they were the least susceptible to microbial adhesion.

The occurrence of corrosion of metal brackets is important, both because of the release of ions and therefore the biocompatibility of these components, and the potential increase in friction as the surface of the bracket slot progressively degrades during treatment [29,30,31,36]. Corrosion resistance is largely dependent on the material of the component and the environmental conditions in which it is placed, i.e., the conditions in the patient’s oral cavity. Nevertheless, the varied surface formation of the bracket can affect the intensity of electrochemical phenomena [29,30,31,36]. In this context, as in the case of the degree of bacterial biofilm adhesion, the lack of homogeneity, in terms of the surface formation of components manufactured by a single manufacturer, makes it impossible to extrapolate the results obtained via in vitro tests to clinical conditions in a fully reliable manner.

All the above considerations are based on the non-contact, optical, indirect assessment of the surface structure. However, it is important to determine whether the adopted methods allow for the assessment of surface roughness or only for a comparative assessment of the type of surface irregularities. Research by Myshkin et al. clearly showed that the fractal dimension analysis allows for a precise determination of the roughness of metal elements—and such were assessed by the authors [37]. In turn, Chappard et al. showed that both texture analysis and fractal dimension analysis clearly correlate with the results of the profilometric test [38]. Therefore, it can be concluded that the results obtained by the authors based on the fractal dimension analysis and the analysis of textures allow for drawing conclusions concerning, not only the comparisons of the unevenness patterns on the surface of the slots of the tested metal brackets, but also the assessment of the roughness of the tested surfaces.

## 5. Conclusions

### Orthodontic Brackets

Mini Sprint brackets presented the most homogeneous and smoothest surface.

Depending on the research method, the remaining brackets may show a high degree of heterogeneity of the bracket slot surface.

Orthodontic brackets can show varying degrees of roughness within brackets produced by the same manufacturer.

The repeatability of the surface structure of the bracket slot can vary from manufacturer to manufacturer.

Manufacturers of orthodontic brackets need to pay attention to maintaining the repeatability of the surface structure of bracket slots.

## 6. Study Limitations

The study was conducted only by means of fractal dimension analysis and texture analysis. The results obtained were not compared with the results of the measurement of friction of individual brackets, the analysis of bacterial adhesion, or the analysis of corrosion phenomena.

The results obtained were not compared with other surface structure analysis methods.

## Figures and Tables

**Figure 1 materials-15-02071-f001:**
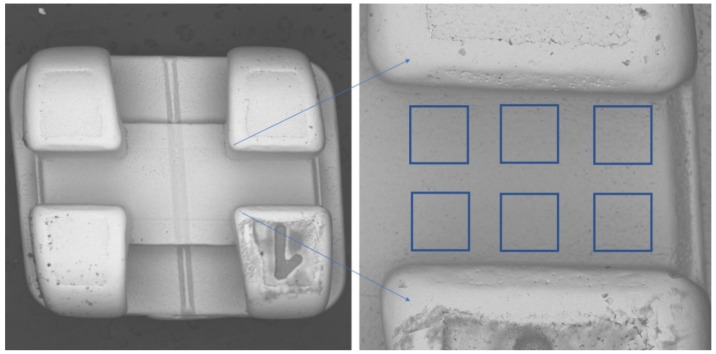
Location of regions of interest (ROIs).

**Figure 2 materials-15-02071-f002:**
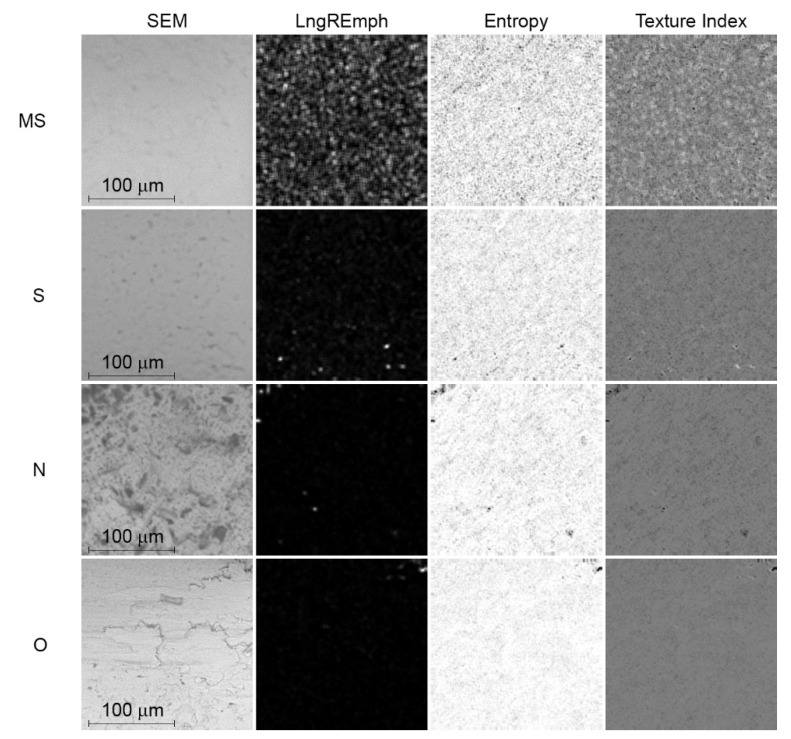
Digital texture analysis of SEM images. Map of the result of LngREmph (from run-length matrix) and Entropy (from co-occurrence matrix) feature calculations. The rightmost column shows the result map of the Texture Index calculation. The LngREmph, Entropy and Texture Index columns represent the intensity tensions of each of these three texture features. The whiter the area, the higher the local intensity of the feature under study.

**Figure 3 materials-15-02071-f003:**
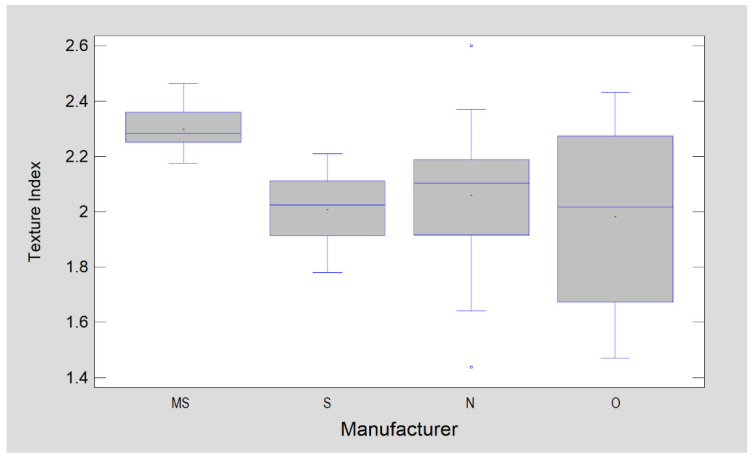
Comparison of orthodontic brackets in terms of surface texture described by Texture Index.

**Table 1 materials-15-02071-t001:** Mean values of the fractal dimension for the tested series of individual slots (MED—mean, SD—standard deviation, 1–4 individual brackets).

Series:	1	2	3	4
	MS			
Med.	1.6426	1.6660	1.6290	1.6346
SD	0.0415	0.0451	0.0090	0.0193
	S			
Med.	1.6700	1.7389	1.6708	1.7194
SD	0.0307	0.0150	0.0147	0.0347
	N			
Med.	1.6383	1.6428	1.6615	1.6423
SD	0.0524	0.0322	0.0457	0.0262
	O			
Med.	1.6704	1.6559	1.6513	1.6426
SD	0.0212	0.0211	0.0222	0.0175

**Table 2 materials-15-02071-t002:** Results of the least significant difference test between the mean value of the fractal dimension (FD) of the slot surface of brackets of different manufacturers.

Manufacturer	FD	*p* < 0.05 (as Compared with)
MS	1.6431 ± 0.0334	S
S	1.6998 ± 0.0389	MS, N, O
N	1.6462 ± 0.0389	S
O	1.6462 ± 0.0218	S

## Data Availability

Data is available from the authors of michal.sarul@umw.edu.pl; kamil.jurczyszyn@umw.edu.pl.

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
