# Peer review of "Surface Evaluation of Orthodontic Brackets Using Texture and Fractal Dimension Analysis"

_materials, 2022, doi:10.3390/ma15062071_

Round 1
Reviewer 1 Report
A very thorough and well documented research on an interesting subject. Although the present paper might present a high degree of interest for readers, certain improvements must be made in order for it to be relevant:
The goals of the present article are not clearly stated in the introduction section. The relevance of the study is not well enough debated in the same section.
References are present in the conclusions section.
The surface analysis was conducted only for new brackets. Since the orthodontic treatment may take several months and even years, these initial surface properties are not relevant for a long time. Surface analysis tests should have been conducted on used brackets as well in order for the results to be more relevant.
The surface of the bracket slot is relevant in relation to the wires but the rest of the bracket's surface is much more relevant when it comes to bacterial adhesion. Was any other part of the bracket studied, besides the slot?
Are there any aspects that can be helpful in everyday practice resulting from the present study?
Please check reference format.
Please provide more recent references (only 5 out of 32 references are less than 5 years old).
Author Response
Dear Reviewer
We would like to thank You for Your insightful review. As we truly believe that it will help us to improve our paper we tried to incorporate all Your remarks in our corrected article.
The goals of the present article are not clearly stated in the introduction section. The relevance of the study is not well enough debated in the same section.
Thank You for this comment
We changed the appropriate paragraph in the introduction and in the beginning of the Materials and methods section
We have also added a relevant sentence in the introduction.
References are present in the conclusions section.
Thank You for this remark. We checked the structure of the article again
The surface analysis was conducted only for new brackets. Since the orthodontic treatment may take several months and even years, these initial surface properties are not relevant for a long time. Surface analysis tests should have been conducted on used brackets as well in order for the results to be more relevant.
Yes, this aspect is extremely important, but it does not fall within the scope of the research objective.
First of all, our goal was to determine how accurate and repeatable the production process of each producer is.
The second major aspect is that the surface of the slots is subject to so many variations during the healing process that it is impossible to obtain a homogeneous interaction with all brackets.
Another aspect is the nature of the study. After many months of exposure to the oral cavity environment, the surface of the brackets may be covered with organic and inorganic deposits. Analysis of fractal dimensions and texture analysis are not, in our opinion, the best method to distinguish an area covered with such sediments from a surface free of sediments. We believed that other methods should be used to evaluate the surfaces of the brackets used.
The surface of the bracket slot is relevant in relation to the wires but the rest of the bracket's surface is much more relevant when it comes to bacterial adhesion. Was any other part of the bracket studied, besides the slot?
Thank You for this comment. Other surface may be more relevant when it comes to bacterial adhesion. However bacterial adhesion was not the only one discussed aspect. For example friction during sliding mechanics is strongly correlated with surface structure of the slot. So the surface of the slot analysis was the objective of this paper. The other thing is that Fractal dimension analysis is much more precise and accurate when performed on the flat surface – so is the slot of the bracket. But we also believe that Your comment shows an important aspect of the studies of brackets surface – we did a necessary addition in Study limitations section.
Are there any aspects that can be helpful in everyday practice resulting from the present study?
Thank You for this comment. Yes we believe that choosing the brackets with most unified and smooth slot surface may be very important in the everyday practice. We showed directly which brackets are the best in this manner – in Conclusion section.
Please check reference format.
Thank You for this comment. Of course we checked it once again and tried to fix all mistakes - thank You.
Please provide more recent references (only 5 out of 32 references are less than 5 years old).
Thank You for this comment. We changed cited articles for the more recent in every case it was possible having in mind the logical and substantive aspect of the work.
Sincerely Yours,
Authors
Reviewer 2 Report
Surface Evaluation of Orthodontic Brackets Using Texture and Fractal Dimension Analysis
The surface morphology of orthodontic brackets is important in shaping the oral cavity environment and the effectiveness of the treatments. A uniform, smooth surface is a sine-qua-non condition for the supplier of orthodontic brackets. In this study, the fractal dimension analysis and texture analysis were performed by 4 selected brackets from three different manufacturers (MiniSprint, Sprint, NuEdge, Orthos SS). The area of each bracket slot was analyzed at 6 predefined points. The smoothest and most uniform and reproducible surface structures were found in the sample by the MiniSprint bracket. The Sprint brackets showed the least homogeneous and least repeatable surface structure.
The surface texture of orthodontic brackets was evaluated using features derived from two groups (run-length matrix and co-occurrence matrix) and texture index.
Selected image texture features based on the entropy and entropy difference from the co-occurrence matrix and long-run emphasis moment from the run-length matrix in the region of interest were calculated for the reference bone and the bone with the collagen scaffold.
- Among all samples, all brackets showed a fractal dimension closest to 1.7, what is the geometric meaning of this average fractal dimension? Is it neither 1D nor 2D?
- What is the size of (ε) box size selected for analysis? There must be a range of optimal values for analysis.
- Definitions and formulations of the co-occurrence matrix, long-run emphasis moment, and the run-length matrix should be provided.
- Please elaborate more details on the calculation of p(i, j) rather than just putting words.
- Any direct correlation between the calculated texture index or fractal dimension to the surface roughness? Authors may need to explicitly state this correlation somewhere in the manuscript.
- The adhesion of pathogenic microorganisms depends, on the degree of surface roughness of the components used in the oral cavity. Can the authors provide some quantitative assessment on this well-known issue? Does this mean what is the range of surface roughness that could promote the growth of micro-organisms?
- Data for statistical analysis, including ANOVA, should be provided as supplementary information. This is important for the credential of numerical results.
- Scale bars (or rulers) are needed for Figure 2 Digital texture analysis of SEM images.
- Please label equation numbers. There is a typo error in the equation of Entropy.
- Grammar and writing need to be improved.
Author Response
We would like to thank You for Your insightful review. As we truly believe that it will help us to improve our paper we tried to incorporate all Your remarks in our corrected article.
Among all samples, all brackets showed a fractal dimension closest to 1.7, what is the geometric meaning of this average fractal dimension? Is it neither 1D nor 2D?
Thank You for this comment. In classical Euclidian geometry we used to know that dimension is an integer value. This value indicates how many variables we need to describe dimension of examined object, for example 0 is dimension of point (0D), to describe segment of line (1D), we need 1 variable – length, flat figures have 2 dimensions (2D) (length, width) and solids have 3 dimensions (length, width, height – 3D). In the fractal geometry a number of dimensions are a rational numbers in range between 0 and 3, for example FD of square is 2, Sierpinski’s carpet equals approximately 1.8928. It means that in infinity scale Sierpinski’s carpet is something between 1 dimension and 2 dimensions shape with tendency to 2D (1.8928 is closer to 2 than 1). Fractal dimension value of images in finite scale (for example microscopic photography) becomes value between 1 and 2. The lower value of FD, the more complex is analyzed shape.
We add this paragraph to Discussion.
What is the size of (ε) box size selected for analysis? There must be a range of optimal values for analysis.
Thank You for this comment.
Image size selected for analysis was 200 x 200 mm. FDA consists of some repeatable steps. For example: at the first step gird size equals 200 mm (dimension of analyzed image, ε=1), in next steps ε is divided by 2 (ε value for following steps: ε=0.5, ε=0.25). In each step difference of pixels bright intensity is calculated in every gird in scale ε. In FracLac software this algorithm of ε calculation is called block series. This option scans a square block within an image using series of grids calculated from the block size. This way is most usable for analysis of pattern which fills whole area of image.
This information was added to the M&M section
Definitions and formulations of the co-occurrence matrix, long-run emphasis moment, and the run-length matrix should be provided.
Thank You for this comment.
Acoording to Haralick, R.M.; Shanmuga, K.; Dinstein, I. Textural features for image classification. IEEE Trans Syst Man Cybern. 1973, 3, 610-621, doi: 10.1109/TSMC.1973.4309314 and Haralick, R.M. Statistical and structural approaches to texture. Proc. IEEE 1979, 67, 786–804, doi:10.1109/PROC.1979.11328
- co-occurrence matrix
The second-order histogram is defined as the co-occurrence matrix hdÆŸ(i,j) [Dash, M.; Liu, H. Feature selection for classification, Intel Data Anal, 1997, 1, (1–4), 131-156, https://doi.org/10.1016/S1088-467X(97)00008-5]. When divided by the total number of neighboring pixels R(d,ÆŸ) in ROI, this matrix becomes the estimate of the joint probability, pdÆŸ(i,j), of two pixels, a distance d apart along a given direction ÆŸ having particular (co-occurring) values i and j. Formally, given the image f(x,y) with a set of Ng discrete intensity levels, the matrix hdÆŸ(i,j) is defined such that its (i,j)th entry is equal to the number of times that
and ,
where .
This yields a square matrix of dimension equal to the number of intensity levels in the image, for each distance d and orientation ÆŸ. In MaZda, the distances d = 1, 2, 3, 4 and 5 pixels with angles ÆŸ = 0°, 45°, 90° and 135° are considered. Reduction of the number of intensity levels (by quantization to fewer levels of intensity) helps increase the speed of computation, with some loss of textural information.
The co-occurrence matrix-derived parameters computed by MaZda are defined by the equations that follow, where µx, µy and ?x, ?y denote the mean and standard deviations of the row and column sums of the co-occurrence matrix, respectively [related to the marginal distributions px(i) and py(j)].
- run-length matrix
Let p(i,j) be the number of times there is a run of length j having grey level i. Let Ng be the number of grey levels and Nr be the number of runs. Definition of the parameter of the run-length matrix p(i,j), as adopted in MaZda, is given below.
3.long-run emphasis moment [source: manual of MaZda program http://www.eletel.p.lodz.pl/programy/mazda/index.php?action=mazda_46]
Please elaborate more details on the calculation of p(i, j) rather than just putting words.
Thank You for this comment.
This calculation follows the assumption made in the 612 pages study by Robert Halalick in 1973 [Haralick RM, Shanmuga K, Dinstein I. Textural features for image classification. IEEE Trans Syst Man Cybern, 1973, 3, 610-621, doi: 10.1109/TSMC.1973.4309314), (see attachment 1)
However, such an elaborate analysis as Haralick's was not applied because directional texture features were not sought. Therefore, data from four directions (00, 450, 900,1350) were averaged. Furthermore, due to the size of the textures [resolution of the analyzed images], d was dropped. Texture features were examined only at a distance of 5 pixels. Thanking you for this useful remark, I add Haralick's 1973 paper to the literature as a source for the method used.
Any direct correlation between the calculated texture index or fractal dimension to the surface roughness? Authors may need to explicitly state this correlation somewhere in the manuscript.
Thank You for this comment.
Of course, this issue definitely needs proper explanation. We have added a corresponding paragraph in the discussion section that relates to the correlation of fractal dimension analysis and texture analysis with profilometry measurement.
The adhesion of pathogenic microorganisms depends, on the degree of surface roughness of the components used in the oral cavity. Can the authors provide some quantitative assessment on this well-known issue? Does this mean what is the range of surface roughness that could promote the growth of micro-organisms?
Thank You for this comment. As it is proven - also in authors` not published yet data - bacteria adhesion depends on many factors and also on terms of experimental methodology. However it is proven that there is strong correlation between roughness of metal objects and formation of bacterial biofilm. We added necessary reference and sentence in Discussion section.
Data for statistical analysis, including ANOVA, should be provided as supplementary information. This is important for the credential of numerical results.
Thank You for this comment. Results of p value for ANOVA has been added as table S1 and S2 in supplementary materials.
Scale bars (or rulers) are needed for Figure 2 Digital texture analysis of SEM images.
Thank You for this comment. Scale bar has been added.
Please label equation numbers.
Thank You for this comment. We added equations numbers.
There is a typo error in the equation of Entropy.
Thank You for this comment. The equation is basing on Haralick, R.M.; Shanmuga, K.; Dinstein, I. Textural features for image classification. IEEE Trans Syst Man Cybern. 1973, 3, 610-621, doi: 10.1109/TSMC.1973.4309314 ( Appendix 1, page 619). I cannot find the typo error. Please forgive me, but I don't know what to correct. The pattern is taken from the MaZda Program Manual and looks in the manuscript to be identical to the one given there:
http://www.eletel.p.lodz.pl/programy/mazda/index.php?action=mazda_46
We added 2 more references:
- Szczypinski, M. Strzelecki, A Materka, A. Klepaczko, MaZda-A software package for image texture analysis, Computer Methods and Programs in Biomedicine, 94(1), 2009, pp 66-76
- Szczypinski, M. Strzelecki, A. Materka, MaZda - a Software for Texture Analysis, Proc. of ISITC 2007, November 23-23, 2007, Republic of Korea, pp. 245-249.
Grammar and writing need to be improved.
Thank You for this comment. It was corrected.
Best regards,
Authors.

Reviewer 3 Report
The paper ”Surface Evaluation of Orthodontic Brackets Using Texture and Fractal Dimension Analysis” can be published in the Materials Journal just after some major revisions.
In my opinion should be revised starting with the introduction section: must be added some paragraphs regarding the influence of elements in production of the brackes; then other paragraph reffering more cleary what other researchers do from the point of view of microstructure and mechanical properties;
Figure 2 is slightly unclear and please improve.
The Conclusion paragraph is to short, please add more text.
Author Response
We would like to thank You for Your insightful review. As we truly believe that it will help us to improve our paper we tried to incorporate all Your remarks in our corrected article.
In my opinion should be revised starting with the introduction section: must be added some paragraphs regarding the influence of elements in production of the brackes; then other paragraph reffering more cleary what other researchers do from the point of view of microstructure and mechanical properties;
Thank You for this comment. We added paragraphs in Introduction and Discussion sections.
Figure 2 is slightly unclear and please improve.
Thank You for this comment. The caption has been elongated twice by adding the explanation: “The LngREmph, Entropy and Texture Index columns represent the intensity tensions of each of these three texture features. The whiter the area, the higher the local intensity of the feature under study.”
The Conclusion paragraph is to short, please add more text.
Thank You for this comment. We expanded the Conclusion section
Best regards,
Authors.
Round 2
Reviewer 1 Report
Dear authors,
The requested modifications and explanations were provided. Thank you for your effort.
Best wishes
Reviewer 2 Report
- The entropy should be ∑i∑jp(i,j)log(p(i,j)) in the original manuscript.
- Equations are not appropriately shown in the revised manuscript, please correct them.
- Please verify grammar and spelling thoroughly.
Reviewer 3 Report
It can be accepted in current form.